

# Physical state of 2-methylbutane-1,2,3,4-tetraol in pure and internally mixed aerosols

Jörn Lessmeier[†1], Hans Peter Dette[†1], Adelheid Godt[*1] and Thomas Koop[*1]

[1]Faculty of Chemistry and Center for Molecular Materials (CM2), Bielefeld University, Universitätsstraße 25, D-33615 Bielefeld, Germany

[†] These authors contributed equally to this work.

*Correspondence to*: Thomas Koop (thomas.koop@uni-bielefeld.de) and Adelheid Godt (godt@uni-bielefeld.de)

**Abstract.** 2-Methylbutane-1,2,3,4-tetraol (hereafter named tetraol) is an important oxidation product of isoprene and can be considered as a marker compound for isoprene-derived secondary organic aerosols (SOAs). Little is known about this compound's physical phase state, although some field observations indicate that isoprene-derived secondary organic aerosols in the tropics tend to be in a liquid rather than a solid state. To gain more knowledge about the possible phase states of tetraol and of tetraol-containing SOA particles, we synthesized tetraol as racemates as well as enantiomerically enriched materials. Subsequently the obtained highly viscous dry liquids were investigated calorimetrically by differential scanning calorimetry revealing subambient glass transitions temperatures $T_g$. We also show that only the diastereomeric isomers differ in their $T_g$ values, albeit only by a few kelvin. We derive the phase diagram of water/tetraol mixtures over the whole tropospheric temperature and humidity range from determining glass transition temperatures and ice melting temperatures of aqueous tetraol mixtures. We also investigated how water diffuses into a sample of dry tetraol. We show that upon water uptake two homogeneous liquid domains form that are separated by a sharp, locally constrained concentration gradient. Finally, we measured the glass transition temperatures of mixtures of tetraol and an important oxidation product of α-pinene-derived SOA: 3-methylbutane-1,2,3-tricaboxylic acid (3-MBTCA). Overall, our results imply a liquid-like state of isoprene-derived SOA particle in the lower troposphere at moderate to high relative humidity, but presumably a semisolid or even glassy state at upper tropospheric conditions, particularly at low relative humidity, thus providing experimental support for recent modelling calculations.



# 1 Introduction

Aerosol particles have been identified as a key factor for answering what drives our climate and, in particular, what is the anthropogenic contribution to climate change (Carslaw et al., 2013; Gordon et al., 2016; IPCC, 2013; Seinfeld et al., 2016). Aerosol particles influence our climate directly through scattering and reflection as well as absorption of sunlight and of
infrared radiation emitted from the earth's surface, and indirectly by acting as nuclei for water and ice cloud formation (Andreae, 1997; Haywood and Boucher, 2000; Lohmann and Feichter, 2005; Pöschl, 2005). These direct and indirect effects are governed by the particles' chemical and physical properties such as their chemical composition and phase state, both of which affect important aerosol processes such as gas-particle partitioning, the rate of surface or bulk chemical reactions, water uptake kinetics, their light-scattering properties, and their ability to act as cloud condensation nuclei or ice nuclei (Baustian et
al., 2013; Berkemeier et al., 2013, 2014; Kidd et al., 2014; Kuwata and Martin, 2012; Martin, 2000; Moise et al., 2015; Perraud et al., 2012; Saukko et al., 2012b; Schill et al., 2014; Shiraiwa et al., 2012; Zarzana et al., 2012). Despite considerable progress in our understanding of atmospheric aerosol processes, aerosols are still considered one of the largest uncertainties when predicting the current and future climate through model calculations (IPCC, 2013). This is especially true for secondary organic aerosol (SOA) particles that originate from the oxidation of biogenic and anthropogenic volatile organic compounds (VOC)
(Hallquist et al., 2009; Jimenez et al., 2009; Kanakidou et al., 2005; Kroll and Seinfeld, 2008; Shrivastava et al., 2017). The great variability in volatile precursors, atmospheric oxidants, and possible reaction pathways leads to a multitude of SOA compounds (Goldstein and Galbally, 2007) thus complicating the prediction of the chemical and physical properties of SOA particles (Donahue et al., 2011; Hallquist et al., 2009; Koop et al., 2011; Kroll et al., 2011; Li et al., 2016).

One of the major uncertainties is the aerosol particles phase state. Atmospheric in-situ measurements of the phase state of
aerosol particles are scarce and often consist of indirect methods such as deducing the phase state from the bouncing behavior of aerosol particles upon collision onto a solid surface (Bateman et al., 2014; Li et al., 2017; Saukko et al., 2012a, 2012b, Virtanen et al., 2010, 2011). This lack of direct in-situ measurements emphasizes the importance of laboratory studies that focus on the phase state and phase changes of representative atmospheric SOA compounds as a function of humidity, temperature and mixing state (Bateman et al., 2014; Dette et al., 2014; Dette and Koop, 2015; Saukko et al., 2012b; Zobrist et
al., 2008). The results of such measurements can be compared to results of atmospheric observations (Bateman et al., 2015; Virtanen et al., 2010) and they can be incorporated into models of atmospheric aerosol phase states (Shiraiwa et al., 2017; Shrivastava et al., 2017).

For a long time SOA particles had been assumed to be always in a liquid phase state at atmospheric conditions (Hallquist et al., 2009; Jimenez et al., 2009; Pankow, 2007). More recently, however, several studies showed that they may indeed occur as
liquids, but can also be present as highly viscous semisolids or even in a solid glassy state, depending on the ambient relative humidity, temperature and precursor substances (Bateman et al., 2015; Koop et al., 2011; Kuwata and Martin, 2012; Mikhailov et al., 2009; Saukko et al., 2012a; Virtanen et al., 2010). For example, Virtanen et al. showed that SOA particles derived from





α-pinene under the moderate temperature and dry humidity conditions over Scandinavian boreal forests are in a semisolid or even glassy state, whereas Bateman et al. showed that particles derived from isoprene in the warm and humid conditions of the Amazonian rain forests are often present in a liquid state. In the study presented here, we focus on the phase change behavior of 2-methylbutane-1,2,3,4-tetraol (hereafter named tetraol), which has been identified as one prominent marker

compound for isoprene oxidation products and together with other polyols make up a significant portion of isoprene derived SOA (Claeys, 2004; Ebben et al., 2014; Isaacman-VanWertz et al., 2016; Nozière et al., 2015a). Besides monoterpenes such as α-pinene, isoprene is one of the most abundant biogenic volatile organic compounds emitted into the atmosphere, especially over tropical forests, where it is oxidized to a variety of compounds (Claeys, 2004; Edney et al., 2005; Guenther et al., 1995, 2012; Martin et al., 2010; Pöschl et al., 2010). Here, we investigated through calorimetric measurements at which temperatures

tetraol transforms from a liquid into a glass, both as a pure substance and in mixtures with water. Furthermore, we investigated whether the presence of stereoisomers affects the phase state of tetraol. This is of particular interest because recent studies have shown that tetraol can occur as an enantiomerically enriched mixture in the atmosphere due to its direct emission by plants that produce an excess of one of the four stereoisomer of tetraol in addition to the atmospheric non-stereoselective oxidation pathway from isoprene (Ebben et al., 2014; González et al., 2011; Nozière et al., 2011). Therefore, we synthesized

enantiomerically enriched samples of all of the four stereoisomers and investigated these samples in addition to the two racemates. The synthesis is described in detail in this paper. Finally, we investigated changes in the glass transition temperature of tetraol when it is mixed with 3-methylbutane-1,2,3-tricarboxylic acid (3-MBTCA), which is a SOA marker compound for α-pinene oxidation products (Dette et al., 2014; Kostenidou et al., 2018; Szmigielski et al., 2007) and was chosen because isoprene and α-pinene are also found in the same air masses (Bateman et al., 2015; Kuhn, 2002) and their oxidation products

in the corresponding SOA.

## 2 Experimental Methods and Materials

### 2.1 Measurement devices

#### 2.1.1 MARBLES

Some samples were transformed into dry glassy samples by way of spray-drying using the MARBLES (metastable aerosol by

low temperature evaporation of solvent) technique that we developed recently (Dette et al., 2014). In short, an aqueous solution of the water soluble compound is atomized into an aerosol, which is subsequently dried in several consecutive diffusion-drying tubes filled with silica gel beads (Silica Gel Orange, Roth). The dried aerosol particles are collected by impaction directly into a DSC sample pan, which is sealed afterwards within a glove box at a relative humidity (RH) of ≤0.1 %.

#### 2.1.2 Differential scanning calorimetry

A differential scanning heat-flow calorimeter (Q100, TA Instruments) with a temperature accuracy of ±0.4 K at heating/cooling rates of 10 K min$^{-1}$ was used. Instrument calibration was performed using 9 different calibration standards at variable heating



and cooling rates between -10 and +10 K min$^{-1}$ (Riechers et al., 2013). For these studies, 3-10 mg of a material was filled into an aluminum pan (see 2.2) and sealed hermetically. The sample was transferred into the DSC and cooled/heated at a rate of 10 K min$^{-1}$ while measuring the heat flow difference between the sample and an air-filled reference pan. The glass transition of a sample is detected as a nearly stepwise change of the heat flow signal. The corresponding glass transition temperature $T_g$ is

determined as the intersection of the extrapolation of the baseline with the line of maximum slope in the heat flow signal, following a convention by Angell (Angell, 2002). This so-called onset is measured at a heating rate of 10 K min$^{-1}$. Because of small ambiguities in the baseline extrapolation and the determination of the point of maximum slope as well as small differences between repeated measurements, we estimate an accuracy in $T_g$ of about ±2 K, despite the calorimeter's principal accuracy of ±0.4 K. The melting of a sample is detected as a peak in the heat flow signal. The melting temperature is determined

as the point of maximum slope in the low temperature part of the melting peak.

**2.1.3 Water activity measurements**

The water activity of several aqueous samples was measured using a commercial water activity meter (Decagon model series 3 TE) at 25 °C. This device measures the dew point of the gas phase above an aqueous solution, from which the corresponding water activity is obtained with an absolute accuracy of ±0.03.

**2.2 Sample preparation**

2-Methylbutane-1,2,3,4-tetraol (tetraol **1**) was obtained as described in section 3.1 and, in more detail, in the supporting information. 3-methylbutane-1,2,3-tricarboxylic acid (3-MBTCA) was prepared as published (Dette et al., 2014). The water used for the preparation of aqueous mixtures was doubly distilled deionized water.

**2.2.1 Pure tetraols**

$(2R,3S)_{65}$-Tetraol and $(2S,3S)_{64}$-tetraol, which were available only in small amounts, were dried by heating to 80 °C for 120 minutes. The dried samples were transferred into a DSC sample pan at room atmosphere. *Rac*-$(2R,3R)$-tetraol, *rac*-$(2R,3S)$-tetraol, $(2S,3R)_{86}$-tetraol, and $(2R,3R)_{97}$-tetraol, which were available in larger amounts, were dried using the spray drying technique MARBLES as described above.

**2.2.2 Two component mixtures**

Mixtures of stereoisomers were prepared by mixing their dilute aqueous solutions in the desired ratio. The mixture was dried with the MARBLES setup. Mixtures of tetraol and 3-MBTCA were prepared in the same manner.



### 2.2.3 Aqueous tetraol samples

Aqueous tetraol samples of particular mass fractions were prepared by weighing vacuum-dried tetraol and water directly into DSC sample pans at the desired ratio under room atmosphere. The pans were sealed immediately after sample preparation and were stored for at least seven days at room temperature prior to DSC analysis to ensure a complete diffusive mixing of the
components within the sample pan, see discussion below.

For the investigation of water uptake, a vacuum-dried tetraol was weighed into a DSC sample pan inside a glove box (≤0.1 % RH). The sample was then quickly transferred into another temperature- and humidity-controlled chamber (see section 3.4, below). This chamber consists of a double-walled glass container, in which the walls and the bottom are cooled by a temperature-controlled circulating water flow, closed with a lid. In the center of the chamber a 15 mm diameter copper rod is
placed vertically as a support for the DSC sample pan. The chamber is filled with a saturated aqueous NaCl solution up to about 1 cm below the upper end of the rod. Therefore, the air above that bath experienced by the sample in the pan is at a relative humidity of ~75%, when the cooling water temperature and, hence, the bath and rod temperature is set to 0.5 °C (Wexler and Hasegawa, 1954). In the experiment described, a sample of tetraol in a DSC sample pan was exposed to these conditions for 250 minutes. Thereafter, the sample pan was sealed immediately, weighed, and then directly transferred into the
DSC for calorimetric measurement.

## 3 Results and Discussion

### 3.1 Synthesis of 2-methylbutane-1,2,3,4-tetraol (tetraol)

For our study, we needed 2-methylbutane-1,2,3,4-tetraol (tetraol) in gram amounts as racemates and as enantiomerically enriched materials of all of its four stereoisomers. Because of their importance in biological and atmospheric research the
syntheses of the stereoisomers of tetraol have been devised already (Cole-Filipiak et al., 2010; Duvold et al., 1997; Ebben et al., 2014; Enders et al., 2007; Fontana et al., 2000; Ghosh et al., 2012; Giner et al., 2002; Hoeffler et al., 2000; Koumbis et al., 2007; Moen et al., 2007; Robinson et al., 2009; Sharma et al., 2008; Urbansky et al., 2004). Among the reported syntheses, the reports of Ghosh et al. (Ghosh et al., 2012), Moen et al., (Moen et al., 2007) and Sharma et al. (Sharma et al., 2008) appeared most attractive to us as they show an access to both diastereomers through the same reaction sequence and give the
four stereoisomers as stereochemically highly homogeneous materials with enantiomeric excesses (ee) of 80-99%. We decided to use the strategy of Moen et al. (Moen et al., 2007) because it requires the least number of steps. It takes up the procedures described by Anthonsen et al. (Anthonsen et al., 1976, 1980) for the preparation of the two diastereomeric tetraols as racemates. In brief (Scheme 1), we started with the prochiral (*E*)-methylbut-2-enedioic acid (mesaconic acid, (*E*)-**1**) and the prochiral (*Z*)-methylbut-2-enedioic acid (citraconic acid, (*Z*)-**1**) which were converted into the diesters (*E*)- and (*Z*)-**2**. S*yn*-dihydroxylation
of these diesters gave the two diastereomeric diols **3** as the racemates *rac*-(2*R*,3*R*)-**3** and *rac*-(2*R*,3*S*)-**4**. Subsequent reduction provided the two diastereomeric tetraols as the racemates *rac*-(2*R*,3*R*)-tetraol and *rac*-(2*R*,3*S*)-tetraol. The enantiomerically





enriched tetraols were obtained through a kinetic resolution at the stage of the diols **3** via enzymatic esterification. For purification of the tetraols, they were temporarily converted into the diacetonides **5**. In the following, details of the individual synthetic steps and on product analysis are given.

**Scheme 1: Synthesis of the racemic and enantiomerically enriched tetraols. NMO indicates N-methylmorpholine-N-oxide. The reaction was performed at room temperature, if no other temperature is given in the scheme.**

One-step esterifications of the diacids (*Z*)- and (*E*)-**1** on a scale of up to 100 g were achieved with ethanol and sulfuric acid as

10   the reagents (Anthonsen et al., 1980; Kar and Argade, 2002; Klimovica et al., 2011; Tripp et al., 2005) and through heating





the reaction mixture to reflux in a Soxhlet extractor with a thimble filled with MgSO₄ for trapping the evolving water. The diesters (*Z*)- and (*E*)-**2** were isolated by washing with base and subsequent distillation.

Following the report of Moen et al., (Moen et al., 2007) *syn*-dihydroxylations of the diesters (*E*)- and (*Z*)-**2** were achieved through applying $K_2OsO_2(OH)_4$ and *N*-methylmorpholine-*N*-oxide with the addition of citric acid in a mixture of *tert*-butanol

and water. The reported work-up had to be modified for running this reaction on a multigram scale, such as starting with 38 g of diester (*E*)-**2** and 107 g of diester (*Z*)-**2**. Moen et al. described the removal of *tert*-butanol by distillation followed by an extraction of the residual aqueous phase with diethyl ether (Moen et al., 2007). Applying this procedure to isolate diol *rac*-(2*R*,3*S*)-**3** on a multigram scale, we found this procedure to be impractical because several litres of diethyl ether would be needed for extraction of this diol due to its low solubility. Therefore, when thin layer chromatography proofed the reaction to

be complete, the organic and aqueous phases were separated and the aqueous phase was extracted with a mixture of diethyl ether and tetrahydrofuran. Filtering of the combined organic phases through silica gel provided diol *rac*-(2*R*,3*R*)-**3**. We did not test the solubility of the diasteromeric diol *rac*-(2*R*,3*S*)-**3** in diethyl ether. We simply applied the same work-up procedure to obtain diol *rac*-(2*R*,3*S*)-**3**, except for substituting the filtration through silica gel with recrystallization.

In order to obtain enantiomerically enriched materials, kinetic chiral resolutions of *rac*-(2*R*,3*S*)-diol **3** and *rac*-(2*R*,3*R*)-diol **3**

were carried out making use of lipase A from *Candida antarctica* (CAL-A). In the following, if the text applies to enantiomerically enriched materials, the figure of *ee* is denoted as a subscript, e. g. (2*S*,3*S*)₉₇-**4** indicates a material consisting of (2*S*,3*S*)-**4** and (2*R*,3*R*)-**4** in a ratio of 98.5 to 1.5 and therefore having a 97% excess of (2*S*,3*S*)-**4**. Likewise (2*S*,3*S*)ₓ-**4** indicates an enantiomerically enriched material with undetermined *ee*. CAL-A catalyses the esterification of the secondary hydroxyl groups of diols **3** with vinyl butanoate and thereby favours the diols (2*R*,3*S*)-**3** and (2*S*,3*S*)-**3** as reaction partners over

their enantiomers (Moen et al., 2007). The attempt of a chiral resolution of *rac*-(2*R*,3*R*)-diol **3** according to the conditions described by Moen et al. (Moen et al., 2007), however on a scale of 3 g instead of 1 g of diol and with CAL-A immobilized on Immobead 150 instead of Novozym 735, failed. No reaction occurred within 3 days. Increasing the temperature from 7 °C to room temperature and adding more CAL-A caused no change within 6 days. However, esterification occurred, when less toluene and more vinylbutanoate were used. Under these conditions, kinetic resolution of diol *rac*-(2*R*,3*R*)-**3** for 44 h at 14 °C

resulted in a conversion of about 44% and gave diol (2*R*,3*R*)₆₄-**3** and butanoate (2*S*,3*S*)₉₇-**4**. Kinetic resolution of the diastereomeric *rac*-(2*R*,3*S*)-diol **3** for 19.5 h under otherwise identical conditions caused a conversion of about 31% and gave diol (2*S*,3*R*)₇-**3** and butanoate (2*R*,3*S*)₈₆-**4**. The conversion was deliberately kept lower in the latter case because of the reported lower enantioselectivity of this reaction (Moen et al., 2007). To obtain diol (2*S*,3*R*)-**3** with a sufficiently high *ee*-value, diol (2*S*,3*R*)₇-**3** was submitted to a kinetic resolution for about 10 days providing diol (2*S*,3*R*)₆₅-**3** and butanoate (2*R*,3*S*)₇₅-**4**. The

reduced solvent volume as compared to the procedure of Moen et al. (Moen et al., 2007) resulted in partial separation of the diols as a second liquid phase. Because of this heterogeneity in phase, the monitoring of the conversion by taking samples for ¹H NMR spectroscopic analysis was not very exact. Nevertheless, this monitoring was sufficiently indicative of when to stop the reaction. Please note that we did not optimise the reaction conditions, neither with respect to solvent volume and amount of vinyl butanoate nor with respect to temperature and reaction time. The butanoates **4** and leftover diols **3** were separated





through column chromatography. The butanoates were coeluted with a compound that was, based on its [1]H and [13]C NMR spectroscopic data (see supplemental information), identified as butanoic acid. The presence of this compound was considered irrelevant because the next step is the reduction of the diols **3** to the tetraols.

To determine the *ee*-values of the diols **3** and butanoates **4**, gas chromatography (GC) and [1]H NMR spectroscopy using (*S*)-
1,1'-bi-2-naphthol as the chiral shift agent were applied (Table S1). NMR spectroscopy was needed because the enantiomeric diols (2*R*,3*S*)-**3** and (2*S*,3*R*)-**3** were inseparable with the available GC equipment. The [1]H NMR signals of the enantiomers were not well separated, therefore deconvolution was applied to allow the determination of the signal intensities. The analysis of (2*R*,3*R*)$_x$-diol **3** revealed that both methods provide about the same *ee*-values (Table S1). Because in the following synthetic steps no reaction takes place at the stereogenic centres, the enantiomeric excesses determined for the diols and butanoates are
identical to those of the tetraols derived thereof.

The reduction of the diols **3** to the tetraols with lithium aluminium hydride proceeded smoothly, however we did not succeed in removing the inorganic salts. Quantitative [1]H NMR spectroscopy revealed a tetraol content in the isolated material of only 70-85 wt%. Moen et al. appear to have faced troubles here, too. Their work-up included three precipitations and the yields ranged from 20-70% (Moen et al., 2007). Our solution to this problem was to convert the tetraols into the diacetonides **5** in
order to mask the highly polar hydroxyl groups and therefore allow standard column chromatography on silica gel to be applied for the removal of the inorganic salts as well as organic side products. The synthesis of the diacetonide (2*S*,3*R*)$_{100}$-**5** has been described (Anthonsen et al., 1980; Proteau et al., 1999). Just as Proteau et al. (Proteau et al., 1999) we used 2,2-dimethoxypropane in the presence of toluene sulfonic acid. Our observations indicate that (2*R*,3*S*)-tetraol and (2*S*,3*R*)-tetraol react slower than their diastereomers, which may simply be the consequence of lower solubility in 2,2-dimethoxypropane, but
also may be caused by the difference in configuration. Because we used an acidic ion exchange resin as the catalyst for demasking, the catalyst could be removed from the tetraols simply by filtration. In this way materials with a tetraol content of 98-99±3 wt% were obtained.

Please note that we synthesised the enantiomerically enriched tetraols before the racemic tetraols. Therefore, the procedures reported for the latter are the more refined, albeit not optimised ones, which we, if we needed more material, would apply to
the enantiomerically enriched materials. The procedure of reduction and work-up to obtain pure tetraol can possibly be slimmed down to (1) hydrolysis of the reduction reaction mixture, neutralisation or mild acidification with sulfuric acid, and removal of solvent and water, (2) reaction with 2,2-dimethyoxypropane, (3) chromatography on silica gel, and (4) acetonide cleavage. We expect the masking as acetonide and subsequent demasking to be applicable also to the isolation of other polyols from mixtures with polar substances.

## 3.2 Glass transition temperatures of the tetraols

The samples available for the calorimetric study are summarized in table 1. All materials - two racemates and four enantiomerically enriched materials - were obtained as highly viscous colorless liquids, in agreement with previous notes of a highly viscous or gel-like state (Ebben et al., 2014). When these liquids were cooled in the bulk state, none of them crystallized



but all exhibited a liquid-to-glass transition. For each of the 6 materials, the glass transition temperature was determined by differential scanning calorimetry. Because of their hygroscopic nature, the samples required drying before the DSC measurements. All materials were dried through heating at reduced pressure. When sufficient material was available, it was spray-dried using the MARBLES technique.

| tetraols | $rac$-(2R,3S) | (2R,3S)$_{65}$ | (2S,3R)$_{86}$ | $rac$-(2R,3R) | (2R,3R)$_{97}$ | (2S,3S)$_{64}$ |
|---|---|---|---|---|---|---|
| $T_g$ / K vacuum-dried | 237 | 242 | 242 | 233 | 237 | 234 |
| $T_g$ / K spray-dried | **245** | | **245** | **238** | **238** | |
| $T_m$ / K | 363 | 352 | 353 | | | |

Table 1. Glass transition temperatures $T_g$ of vacuum-dried and spray-dried racemic and enantiomerically enriched tetraols and melting temperatures $T_m$ for crystallized samples. The enantiomeric excess is given as subscript. The spray-dried samples were handled and sealed in a dry glove box before DSC measurement and, therefore, provide the most accurate $T_g$ values, which is why 10 they are given in boldface.

For racemic and enantiomerically enriched tetraols, $T_g$ values for vacuum-dried and spray-dried samples were obtained, see table 1. Vacuum-dried samples always showed slightly lower $T_g$ values than spray-dried samples. The most likely reason for this difference is that vacuum-dried samples were transferred into DSC sample pans and weighed at room atmosphere. 15 Therefore, most probably, uptake of trace amounts of water occurred, which acted as a plasticizer thus reducing the $T_g$ values of these samples. The amount of water taken up by the sample can vary from experiment to experiment, for example caused by slightly varying times needed for the sample preparation. This explains as well why there is a difference in the $T_g$ of (2R,3R)$_{97}$-tetraol and (2S,3S)$_{64}$-tetraol, which should otherwise be identical due to their enantiomeric nature. We conclude that the $T_g$ values of spray-dried samples handled and sealed in a dry glove box are much less affected by traces of water and, thus, 20 are more reliable (and therefore written boldfaced in table 1). These measured $T_g$ values of 238 K and 245 K are considerably lower than the value of 287 K ± 12 K predicted by Berkemeier et al. (Berkemeier et al., 2014) which was based on a structure-property relationship modelling approach. This difference emphasizes the importance of laboratory measurements. While structure-property relationship predictions can often provide reasonable estimates for average phase transition temperatures of a large number of substances, they may show significant deviations on an individual compound basis.

25 The spray-dried samples showed no difference in $T_g$ between $rac$-(2R,3S)-tetraol and (2S,3R)$_{86}$-tetraol and between $rac$-(2R,3R)-tetraol and (2R,3R)$_{97}$-tetraol. Due to lack of sufficient amount of material, spray-drying was possible for only one enantiomer of each diastereomer. It is to be expected, that the $T_g$ values of enantiomers are equal because enantiomers share





the same physical properties (Jabrane et al., 1995). Accordingly, our data strongly suggest that there is no difference between the $T_g$ values of the enantiomers and their racemic mixtures of both diastereomers. There is, however, a difference of 7 K between the $T_g$ values of the two diastereomers. This difference was further investigated by mixing the racemates in different mass ratios in an aqueous solution and subsequently spray-drying them with the MARBLES technique. The $T_g$ values of each

of these mixtures were determined with DSC measurements. The resulting $T_g$ values show a nearly linear relationship with mass ratio, as shown in figure 1.

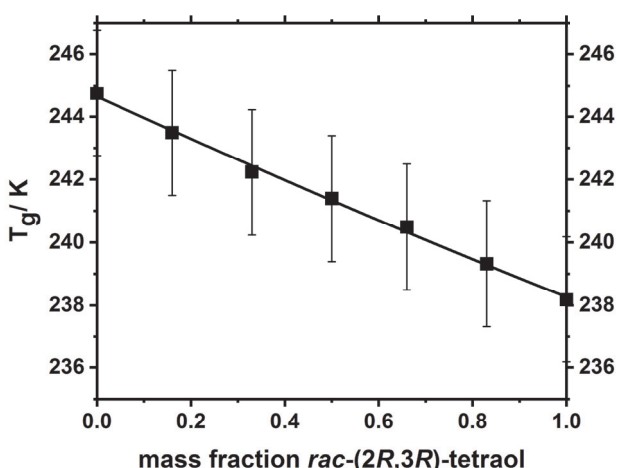

**Figure 1. Glass transition temperatures of mixtures of rac-(2R,3S)-tetraol and rac-(2R,3R)-tetraol. The $T_g$ values show a nearly**
**linear relationship with a Gordon-Taylor constant k_GT of 0.92.**

A Gordon-Taylor-fit (Gordon and Taylor, 1952) of the $T_g$ data (Eq.1) results in a Gordon-Taylor constant $k_{GT}$ of $0.92 \pm 0.09$, implying a nearly linear relationship (i.e. $k_{GT} = 1$), which is expected for chemically similar compounds. In general, the Gordon-Taylor-equation is used to describe the glass transition temperature $T_g$ of binary mixtures, with $w_1$ and $w_2$ as their

respective mass fractions, $T_{g1}$ and $T_{g2}$ as the glass transition temperatures of the pure compounds and an empirical fitting constant $k_{GT}$.

$$T_g = \frac{w_1 T_{g1} + \frac{1}{k_{GT}} w_2 T_{g2}}{w_1 + \frac{1}{k_{GT}} w_2} , \qquad (1)$$

While preparing DSC-samples of rac-(2R,3S)-tetraol, the bulk sample crystallized, whereas none of the other samples crystallized, even during weeks of storage. The crystals of rac-(2R,3S)-tetraol were used as seeds for triggering the



crystallization of the corresponding enantiomers. We note that all attempts of seeding the diastereomers with any of the crystallized compounds were unsuccessful. This may be due to different crystal structures of the two diastereomers. The melting points $T_m$ of the crystallized compounds were also determined, see table 1, and the corresponding ratios $T_g/T_m$ are very close to the value of 0.7 derived for a large number of molecular compounds (Koop et al., 2011). We note that the racemate

*rac*-(2*R*,3*S*)-tetraol shows a $T_m$ significantly higher than those of the individual enantiomers which indicates the formation of a racemic crystal, i.e. a crystal that contains the two enantiomers in a 1:1 ratio in the elementary cell (Roozeboom, 1899).

**3.3 Phase diagrams of mixtures of tetraol with water**

Dry tetraol does not exist in the atmosphere because water is always present. Therefore, it is important to investigate possible phase states and transitions of tetraol/water mixtures as a function of temperature and water content. For example, how does

the tetraol glass transition change with water content, and how strongly is the ice melting point reduced as a function of tetraol mass fraction? Because tetraol and water are totally miscible over the entire concentration range, it is possible to investigate mixtures at any mass ratio between zero and one. We prepared nine different mixtures by weighing appropriate amounts of racemic tetraols and water directly into DSC sample pans, which were sealed immediately. Thereafter, the samples were equilibrated at room temperature for at least 7 days to ensure complete diffusive mixing of the two components. That a

homogeneous mixture was reached within this time period was proven by DSC measurements which revealed only a single glass transition (see section 3.4 below). Figure 2 shows for both racemates partial phase diagrams of tetraol/water mixtures, constructed from the measured phase transition temperatures. The diagrams include the glass transition temperatures $T_g$ (black squares), the ice melting points $T_m$ (red circles), and the glass transition temperature of the maximally freeze-concentrated solution $T_g$' (blue triangles). Note that it is common to plot the $T_g$' values at the overall mass ratio of the samples before ice

formation. The actual mass ratio of the maximally freeze-concentrated solution after the water partly crystallized can be derived by determining the mean of the $T_g$' values, plotting this mean as a horizontal line and obtaining its intersection with the glass transition curve from the Gordon-Taylor-fit (blue star).



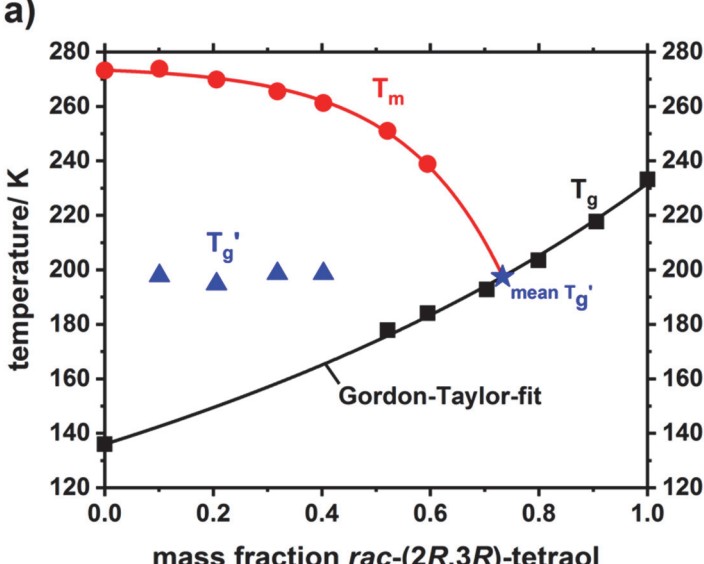

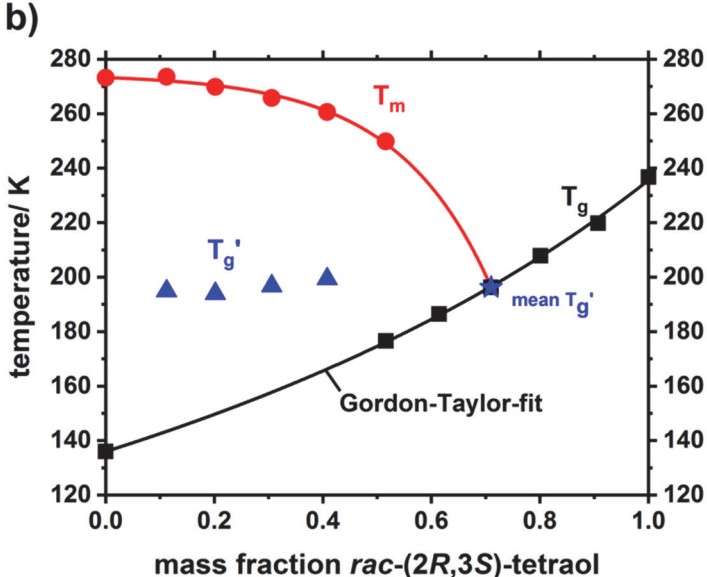

**Figure 2. Partial phase diagram of mixtures of rac-(2R,3R)-tetraol with water (a) and rac-(2R,3S)-tetraol with water (b). Shown are the ice melting temperatures $T_m$ (red circles), glass transition temperatures $T_g$ (black squares) and maximally freeze-concentrated glass transition temperatures $T_g$' (blue triangles). The melting temperatures are fitted with an exponential function and the glass transition temperatures are fitted with the Gordon-Taylor equation yielding a Gordon-Taylor constant of $k_{GT} = 1.53 \pm 0.08$ (a) and $k_{GT} = 1.58 \pm 0.05$ (b), respectivly. The $T_g$' values are plotted at the overall mass fraction of the solution before cooling. The composition of the amorphous fraction of a sample after the water partly crystallized is depicted as the mean $T_g$' shifted towards the Gordon-Taylor fit (blue star).**



The experimental glass transition temperatures were fitted with the Gordon-Taylor equation resulting in a $k_{GT}$ of $1.53 \pm 0.08$ for *rac*-(2*R*,3*R*)-tetraol (figure 2a). The preparation of the tetraol/water mixtures could be done under room atmosphere conditions only, thus, traces of water may have been taken up by the vacuum-dried tetraol during this preparation. For this reason, the tetraol mass fraction calculated for each tetraol/water mixture may be a little too high. Based on the data obtained

for vacuum-dried and spray-dried samples (table 1), the maximum error in the water mass fraction is estimated to be 0.03 for (a) and 0.05 for (b), with smaller errors at smaller tetraol weight fractions. For that reason, we used the $T_g$ values of the vacuum-dried tetraols for the Gordon-Taylor analysis to allow for a better comparability.

The partial phase diagram of *rac*-(2*R*,3*S*)-tetraol (figure 2b) is very similar to that of the diastereomeric racemate. The glass transition curve fitting results in a $k_{GT}$ of $1.58 \pm 0.05$. The enantiomerically enriched materials (2*S*,3*R*)₈₆-tetraol and (2*R*,3*R*)₉₇-

tetraol gave identical phase diagrams as their corresponding racemates within experimental uncertainty and are therefore not shown.

From the two phase diagrams in figure 2 some conclusions can be drawn regarding the atmospheric phase state of tetraols. Aerosol particles that consist predominantly of racemic or enantiomerically enriched tetraol are likely to be amorphous rather than crystalline. At low water mass fractions, i.e. at low atmospheric relative humidity, the particles may be in a glassy or a

highly viscous liquid state at the low temperatures of the upper troposphere. At medium to high water mass fractions, corresponding to moderate or high relative humidity, two scenarios are possible. First, the particles may be in a homogeneous, highly viscous liquid state. Second, when water crystallizes, they are likely to be in a heterogeneous state, consisting of a crystalline ice core that is surrounded by a tetraol-rich shell. This shell would be in a liquid state above the glass transition temperature of the maximally freeze-concentrated tetraol solution at about $T_g' \approx 200$ K for both diastereomers, i.e. over nearly

the entire tropospheric temperature range.

In order to apply the above data better to atmospheric conditions, it is helpful to plot the glass transition temperatures as a function of water activity ($a_w$) rather than mass fraction, because for particles in equilibrium with the surrounding air, $a_w$ directly corresponds to the ambient relative humidity. Therefore, we measured the water activity of tetraol/water mixtures with mass fractions ranging from 0 to 1 at a temperature of 298.15 K. The resulting data are shown in figure 3 together with a

parameterization curve derived by fitting the data according to an approach developed previously (Zobrist et al., 2008). As expected, the water mass ratio dependence of $a_w$ is practically the same for both racemates within experimental uncertainty.





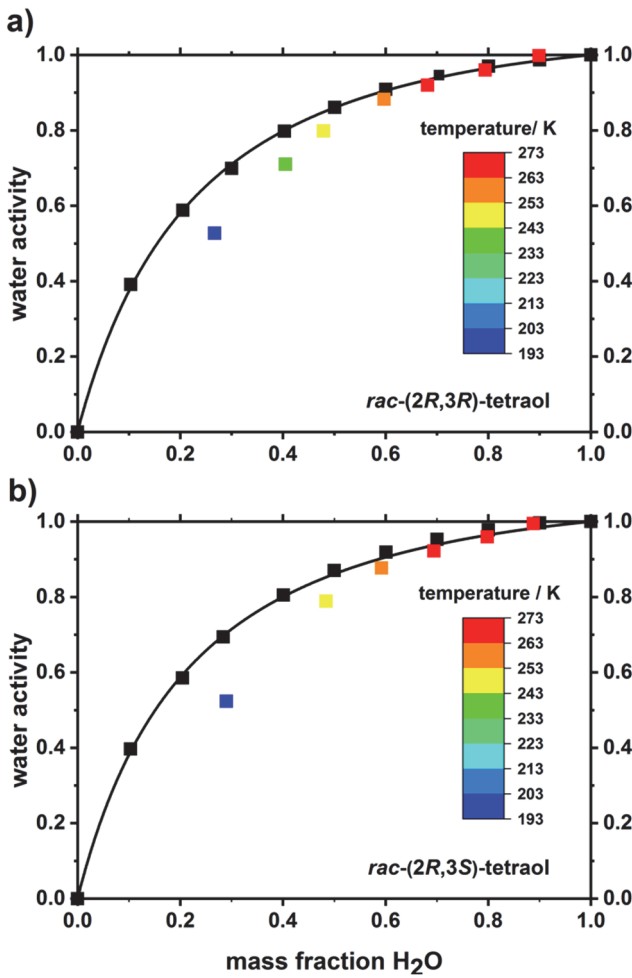

**Figure 3. Measured water activity of mixtures of (a) rac-(2R,3R)-tetraol with water and (b) rac-(2R,3S)-tetraol with water at 298.15 K (black squares) and at their ice melting points (colored squares). The curves are fits to the data: for details see main text.**

5   We further measured ice melting points for the tetraol/water mixtures and calculated the water activity of a corresponding liquid solution at this temperature according to Koop and Zobrist (Koop and Zobrist, 2009). With these data we were able to describe the temperature dependence of $a_w$ as a function of tetraol concentration. By this procedure, the glass transition curve of figure 2 can be transferred from the mass fraction into the water activity regime. The result of this analysis is given in figure 4, where the grey shaded area indicates the temperature range at which an aqueous tetraol aerosol particle most likely

10  transitions from a liquid to a glass upon cooling at a certain relative humidity (RH). The relatively wide range of the glass transition region (3σ; grey shaded area) includes uncertainties in the Gordon-Taylor fits of figure 2 as well as those of the temperature dependence of $a_w$ stemming from two slightly different fitting procedures. (For more details see supporting information.) Also shown in figure 4 is the water activity-dependent ice melting point curve (red line) as well as the homogeneous ice nucleation temperature curve (black line) (Koop et al., 2000; Koop and Zobrist, 2009).



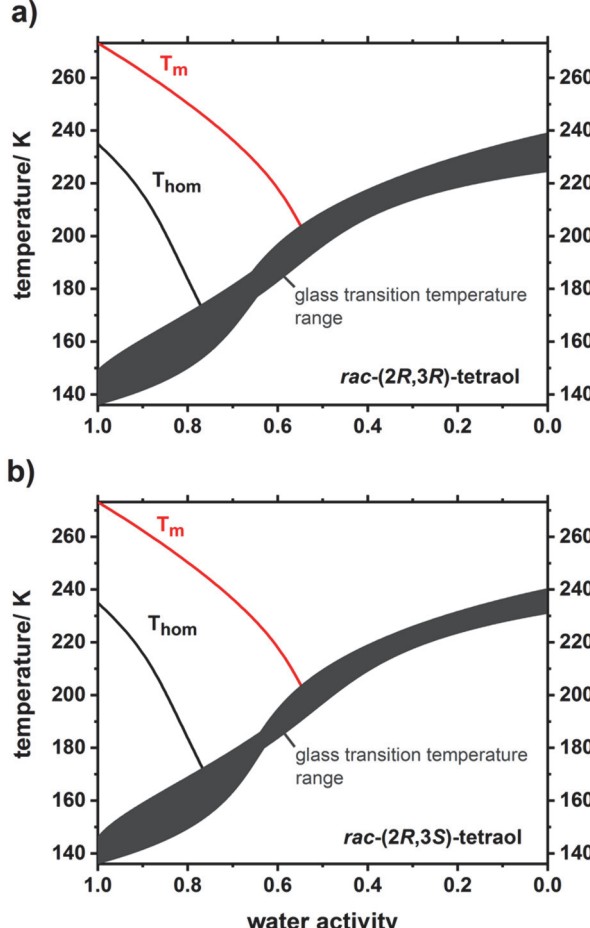

**Figure 4. Partial phase diagrams of mixtures of rac-(2R,3R)-tetraol with water (a) and rac-(2R,3S)-tetraol with water (b) plotted as a function of water activity. In addition to the ice melting point line ($T_m$; red line) the homogeneous ice nucleation temperatures ($T_{hom}$; black line) are shown. The dark grey area indicates the glass transition temperatures and their uncertainty; see text for details.**

The phase diagrams shown in Figure 4 now allow a more detailed comparison of our experimental laboratory results with the findings from field measurements of Bateman et al. 2015 about the phase state of SOA particles in the Amazonian rain forest. They stated that said particles were primarily liquid at least near the surface where temperature was almost always above 296 K and relative humidity was above 80% (i.e. particle water activity is above 0.8) about 70% and 80% of the time in the dry and wet season, respectively (Bateman et al., 2015). Our measurements show that tetraol, as a representative compound for SOA substances in the Amazonian rainforest, is indeed always liquid above 273 K at high and even at low relative humidity. Bateman et al. further observed that the relative humidity dropped to values of ~50% during some hot afternoons in the dry season. They speculated whether these conditions may lead to semisolid or solid particles, although they deemed this not to be





very likely owing to the high temperatures of 300-310 K during these low RH events. Our measurements show that, at least for pure tetraol aerosol particles, the temperature would indeed have to be below 210K at a RH of 50% in order for solid SOA particles to form. Our measurements thus imply that even semisolid SOA particles are unlikely at the observed low RH conditions in the dry season of the Amazonian rain forest.

**3.4 Water uptake by the tetraol**

Tetraol is a hygroscopic compound that takes up water from the ambient air, see discussion above. The kinetics and mechanism of water uptake by hygroscopic aerosol particles is currently an intensely investigated subject. Different ideas on how water uptake into and water diffusion within particles occur are studied experimentally or are being described using different types of modelling approaches (Berkemeier et al., 2013, 2014, Hosny et al., 2013, 2016; Mikhailov et al., 2009; Rothfuss et al.,

2018; Zaveri et al., 2014; Zobrist et al., 2011). It has been shown that particularly in semisolid and glassy matrices water diffusion can be strongly inhibited, thereby leading to concentration gradients within the particle matrix. (Berkemeier et al., 2014; Hosny et al., 2016; Zobrist et al., 2011) Moreover, in the classical case where the diffusion coefficient of the diffusing tracer molecule (here water) is constant throughout the matrix, the time-dependent concentration profile can be calculated assuming Fickian diffusion, thus resulting in relatively shallow exponential concentration gradients. The situation changes

dramatically, when the diffusion coefficient of the tracer is concentration dependent. For example, water acts as a plasticizer in many matrices, including tetraol, thus reducing their viscosity and, thereby, increasing its own diffusion coefficient. As a result, during the water uptake of a particle a sharp transition line with a steep concentration gradient develops between two nearly homogeneous domains, a pure organic core with very low water diffusivity and a well-mixed organic/water shell with high water diffusivity (Berkemeier et al., 2014; Hosny et al., 2016; Mikhailov et al., 2009; Zobrist et al., 2011). As time

progresses, this transition line moves towards the particle's center until finally one homogeneous liquid particle is formed (see figure 5).

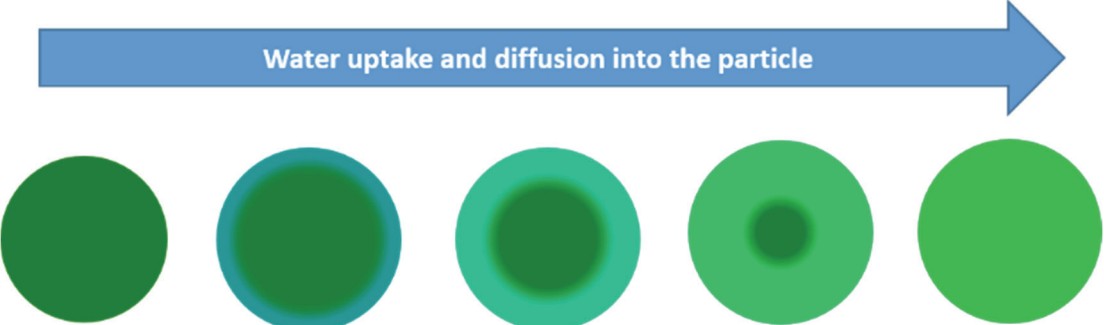

**Figure 5. Schematics of water uptake and diffusion into an aerosol particle with a sharp transition line distinguishing two**
**homogeneous domains during diffusion: the dark green organic core and the light green organic/water shell.**





In the same manner, water diffuses into a macroscopic tetraol sample placed inside a DSC pan when the sample is exposed to a humid atmosphere before sealing the pan (see figure 6 and 7a). In some preliminary DSC measurements this process was observed to occur when samples were exposed to room atmosphere for more than 10 minutes before the samples were sealed: instead of a single glass transition step two distinguishable glass transition steps were revealed in the thermogram, indicating

the presence of two homogeneous but distinguishable compositional domains within the sample. To illustrate this fact, we performed the following experiment. We filled *rac*-(2*R*,3*R*)-tetraol into a DSC sample pan and placed the pan into an environmental chamber on top of a copper rod partly immersed in a tempered saturated aqueous sodium chloride solution (figure 6). The relative humidity over such a solution is practically independent of temperature and amounts to 75% RH. The temperature of the bath and, therefore, also that of the material in the DSC sample pan was set to 0.5 °C, i.e. a temperature at

which the diffusion of water into the sample is slow.

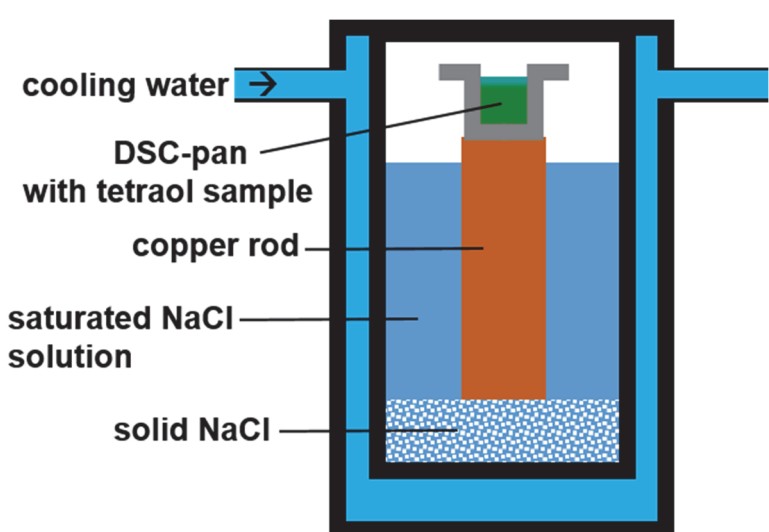

**Figure 6. Sketch of the environmental chamber used in this study to expose rac-(2R,3R)-tetraol to an atmosphere with 75% RH.**
**The sample is placed on top of a copper rod immersed in a saturated aqueous NaCl solution inside a temperature-controlled chamber set to 0.5 °C.**

After 250 minutes of exposure time the DSC pan was taken out of the chamber, immediately sealed, weighed and then placed into the DSC. Thereafter, the sample underwent six cooling/heating cycles at rates of 10 K min⁻¹. In each of the first four cycles

the sample was first cooled to 153 K and subsequently heated to 303 K with a 20 min holding time at this latter temperature. After the 5ᵗʰ cycle, the sample was heated to 353 K and held at this temperature for 20 minutes to ensure complete diffusive mixing, because diffusion is significantly faster at higher temperature.



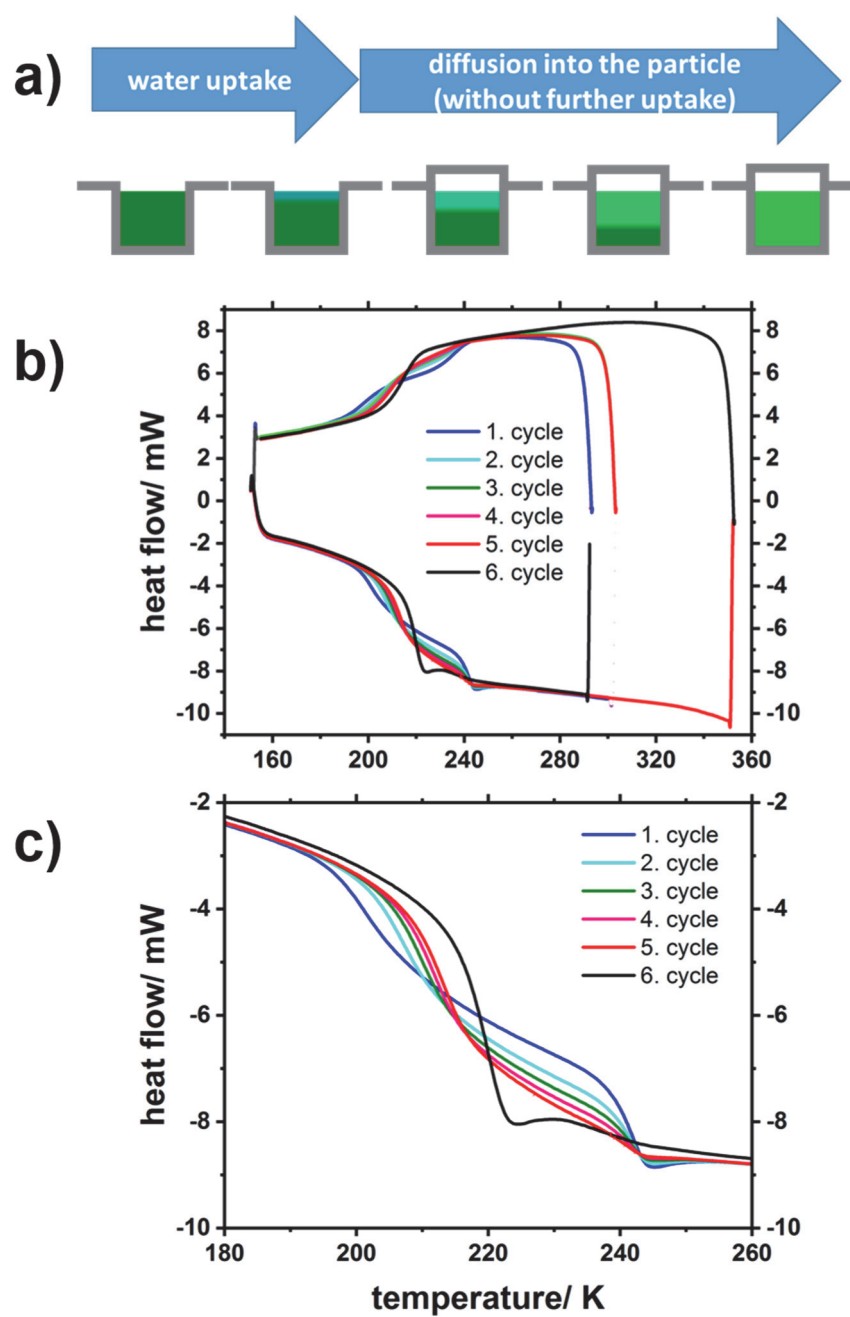

**Figure 7. (a)** Illustration of water uptake by a hygroscopic substance and subsequent water diffusion within it for a sample inside a DSC sample pan. **(b/c)** DSC thermograms showing several consecutive cooling/heating cycles of wetted rac-(2R,3R)-tetraol (b), with a close-up view of the glass transition region (c). After each of the first 4 cycles the temperature was held constant at 303.15 K for 20 minutes before the next cycle was initiated. After the 5th cycle the temperature was raised to 353.15 K and held constant for 20 minutes before the last 6th cycle was initiated.



Figure 7b shows an overview of the measured thermograms, while panel c is a close up of the glass transition temperature range. The first cycle (blue line) reveals that two glass transitions occur in the sample both upon cooling and upon heating, one at about 197 K and another at about 239 K. The latter, higher $T_g$ can be assigned to the bottom domain of the sample consisting of only *rac*-(2*R*,3*R*)-tetraol (see dark green domain in figure 7a), because its glass transition occurs at the same

temperature that was measured for spray-dried *rac*-(2*R*,3*R*)-tetraol within the measurements uncertainty. The lower $T_g$ can be assigned to the upper, water-containing domain of the sample (figure 7a). The glass transition is at lower temperature because water ($T_g$ of 136 K) acts as a plasticizer as discussed above in the analysis of Figs. 2 and 4. In the subsequent cooling and heating cycles changes can be observed for each of the two glass transitions. First, the temperature of the lower $T_g$ moves to higher temperature from one cycle to the next and concomitantly its step height (i.e. the magnitude of the signal) becomes

larger. At the same time, the temperature of the higher $T_g$ does not change from one cycle to the next (238.5 K for the first cycle and 238.6 K for the 5[th] cycle), but its step height decreases with every cycle. These findings are in in full agreement with the anticipated behavior of the sample. While the water diffusion gradient transition line moves deeper into the sample with time, the tetraol core becomes smaller and the aqueous tetraol shell becomes larger (see figure 7a), which explains the development of the $T_g$ step heights. During this process, the total amount of water is constant as the DSC sample pan is sealed.

Therefore, the concentration of water in the growing aqueous shell becomes smaller, leading to an increase in tetraol concentration and, thus, an increase of the lower $T_g$. Finally, after five cycles, the sample was heated to 353 K and held at that temperature for 20 minutes to speed up the mixing process. Indeed, the sample fully equilibrated into one homogeneous phase, as can be seen from the fact that only one single glass transition step at 215 K is observed in the final 6[th] cooling and heating cycle (black lines in figures 7b and c). Because the sample was weighed before and after water uptake, the overall water mass

fraction was known. With this information the anticipated $T_g$ of this mixture could be inferred from the Gordon-Taylor fit shown in figure 2. The measured $T_g$ and the one derived from the Gordon-Taylor fit both amount to 215 K.

### 3.5 Glass transition temperatures of mixtures of tetraol with 3-MBTCA

In the atmosphere, it is very unlikely that aerosol particles consist of only one compound such as tetraol. Instead, atmospheric aerosol particles, and in particular (secondary) organic aerosol particles, predominantly consist of complex mixtures of

compounds, depending upon their precursor compounds and their state of oxidation (Decesari et al., 2006; Hallquist et al., 2009; Jimenez et al., 2009; Kanakidou et al., 2005; Kroll and Seinfeld, 2008; Li et al., 2016; Nozière et al., 2015b; Rogge et al., 1993). Another important representative for secondary organic aerosols is 3-methylbutane-1,2,3-tricarboxylic acid (3-MBTCA). 3-MBTCA is an oxidation product of α-pinene and is often used as a marker compound (Donahue et al., 2012; Szmigielski et al., 2007; Yasmeen et al., 2011). The tetraol studied above is a marker compound for isoprene SOA. Both α-

pinene and isoprene are believed to be the most abundant biogenic VOC precursors in tropical forests such as in Amazonia (Bateman et al., 2015). Hence, we further investigated how the glass transition temperatures of tetraol and 3-MBTCA changes when these compounds are mixed. We used the MARBLES setup to spray-dry aqueous mixtures of 3-MBTCA and *rac*-



(2R,3S)-tetraol with different mass ratios. The $T_g$ values of the dried samples obtained from DSC experiments are shown in figure 8.

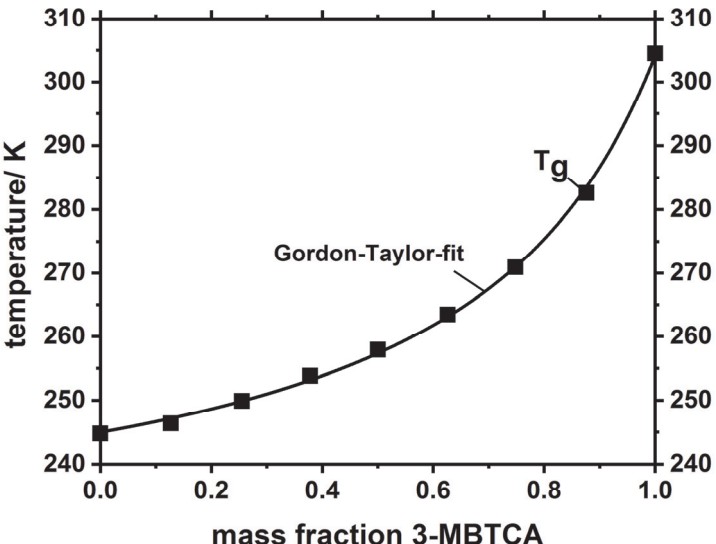

**Figure 8. Glass transition temperatures (points) of spray-dried mixtures of rac-(2R,3S)-tetraol and 3-MBTCA. The black line represents the Gordon-Taylor fit to the data.**

These $T_g$ values can be described very well with the Gordon-Taylor equation and a corresponding value of $k_{GT} = 3.79 \pm 0.15$, underpinning the usefulness of the Gordon-Taylor approach for describing glass transitions even in SOA compound mixtures derived from different precursor gases. A value of $k_{GT}$ of 3.79 implies a relatively strong decrease in $T_g$ even at small to moderate tetraol mass fractions, e.g. from 306 K in pure 3-MBTCA to about 270 K at a 3-MBTCA mass fraction of 0.75. The $T_g$ of aerosol particles with this composition would be even lower when water is taken up, suggesting that SOA particles containing predominantly isoprene and α-pinene oxidation products are in a liquid state at atmospheric conditions in the boundary layer of a tropical forest, in agreement with in-situ observations (Bateman et al., 2015) and global modelling approaches (Shiraiwa et al., 2017). Such aerosol particles may, however, transform into a glassy state in the upper troposphere, i.e. at low temperature and low to moderate relative humidity.

## 4. Summary and Conclusion

We described the syntheses of 2-methylbutane-1,2,3,4-tetraol in form of the two possible racemates and of enantiomerically enriched materials with 65-97% enantiomeric excess in detail with a focus on the isolation of the compounds. A simple




purification of the tetraols via their diacetonides is introduced. We investigated the phase states of the individual stereoisomers of the tetraol, their racemic mixtures, and their mixtures with water. All stereoisomers were obtained as a viscous liquid at room temperature. *Rac*-(2*R*,3*S*)-tetraol and *rac*-(2*R*,3*R*)-tetraol transitioned from a liquid to a glass upon cooling at about 238 K and 245 K, respectively. The glass transition temperatures $T_g$ for the enantiomerically enriched tetraols were found to be

identical with those of the corresponding racemates. The $T_g$ values of mixtures of the two racemates show a small and nearly linear dependence with mass ratio of the racemates.

We also investigated the water uptake and diffusion into a dry tetraol bulk sample by monitoring the sample mass and the glass transition temperature via DSC measurements. Water uptake and diffusion was slow, proceeding on the time scale of hours, just as expected for a semisolid bulk sample. At first two distinct amorphous domains form, which were distinguishable by

their individual glass transition temperatures. The equilibration of the two domains into one homogeneous liquid phase was followed by monitoring the magnitude and temperature of the two glass transitions as a function of time.

We further determined atmospherically relevant phase transition temperatures in tetraol/water mixtures over the entire mass fraction range including ice melting and glass transition temperatures. By determining the water activity of tetraol/water mixtures we were able to construct a phase diagram of these mixtures covering the entire tropospheric temperature and

humidity range. These phase diagrams imply that aerosol particles consisting predominantly of tetraol are most likely in a liquid or highly viscous semisolid state in the atmosphere, and not in a glassy or crystalline solid state. Glass transition measurements of tetraol/3-MBTCA mixtures suggest that SOA particles containing oxidation products of isoprene as well as oxidation products of alpha-pinene are liquid at temperatures of the lower troposphere, but may be semisolid or even glassy at upper tropospheric conditions, particularly at low relative humidity. Our data support using the Gordon-Taylor approach for

describing $T_g$ values of SOA compound mixtures (Shiraiwa et al., 2017), even when their components are rather dissimilar in chemical structure as is the case when they derive from different types of volatile precursors.

## AUTHOR INFORMATION

Corresponding Authors

*T. Koop. E-mail: thomas.koop@uni-bielefeld.de

*A. Godt. E-mail: godt@uni-bielefeld.de.

**Author Contributions**

The study was conceived by all authors, the synthesis was planned and performed by J.L. and A.G., the physical experiments were performed by J.L. and H.P.D., results were analyzed by J.L. and H.P.D. and discussed by all authors. The paper was written with contributions from all authors.



## Acknowledgements

We acknowledge support for the Article Processing Charge by the Deutsche Forschungsgemeinschaft and the Open Access Publication Fund of Bielefeld University. We thank Florian Uthoff for advice at the very early stages of applying enzymatic racemate resolution.

5 **Notes**

The authors declare no competing financial interest.

**Abbreviations**

| | |
|---|---|
| $T_g$ | glass transition temperature |
| tetraol | 2-methylbutane-1,2,3,4-tetraol |
| 10 3-MBTCA | 3-methylbutane-1,2,3-tricarboxylic acid |
| DSC | differential scanning calorimeter |
| IPCC | Intergovernmental Panel on Climate Change |
| VOC | Volatile Organic Compound |
| SOA | Secondary Organic Aerosol |
| 15 MARBLES | Metastable Aerosol By Low Temperature Evaporation of Solvent |
| $T_m$ | melting temperature |
| $T_{hom}$ | homogeneous nucleation temperature |
| $k_{GT}$ | Gordon-Taylor-constant |
| $T_g$' | glass transition temperature of a maximally freeze-concentrated solution |
| 20 $a_w$ | water activity |
| RH | relative humidity |

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
