# Peer review of "Physical state of 2-methylbutane-1,2,3,4-tetraol in pure and internally mixed aerosols"

_Atmospheric Chemistry and Physics, 2018_

## Referee Comment (RC1) · Anonymous Referee #1 · 23 Aug 2018

This manuscript investigated the phase state of tetraol, which is an important secondary organic aerosol component. The authors synthesized teraol and studied the glass transition temperatures (Tg) of tetroal in pure and mixed particles with another important oxidation product, 3-MBTCA. This manuscript derived the phase diagram of water/tetraol mixtures at the atmospheric relevant conditions and determined the water activity of water/tetraol mixtures as a function of temperature and solute mass concentration. This study provides a set of valuable data for the phase state important SOA at different temperature and humidity. Due to the lack of experience on the synthesis, I was not able to provide assessment on this part. Besides that, the experimental methods are valid and the scientific approach and discussion are sound. The paper is well written and organized. I recommend it for publication for a minor revision. Please see

the following comments which the authors may want to consider in the revision.

Minor comments:

1. P11, L17-22, It is not trivial to understand the Tg' with currently short description. It will be useful to provide details how the Tg' was determined. It could be also useful to provide measured average values instead of Tg' is about 200 K.

2. P13, L2-11, How does the Tg look like when using Tg(dry) from the spray-dried samples? It is suggested to derived Tg using both Tg(dry) from vacuum-dried and spray-dried samples, or provide uncertainties in Figure 2. Does the uncertainty in kGT cover this variation? L5-6, the maximum error in the water mass fraction is estimated to be 0.03 (a) and 0.05(b), do you mean in figure 2(a) and 2(b)?

3. P14, L8-13, It will be great to provide the fitting parameters that could be directly used by the readers.

4. P16 section 3.4, is it possible to estimate the diffusion coefficient of water vapor using this water uptake/DSC experiment?

---

## Referee Comment (RC2) · Anonymous Referee #2 · 27 Aug 2018

The authors describe the physical phase states of 2-Methylbutane-1,2,3,4-tetraol and mixtures of tetrol which can be a marker for isoprene-derived SOA and $\alpha$-pinene-derived SOA. The results provide new/additional insight for the phase state of SOA particles which is still unknown in the atmospheric chemistry community. This manuscript is clear, concise, and well-written. I recommend this manuscript for publication in ACP. I have several comments that the authors should consider prior to publication.

1. Page 2, line 19 – page 3, line 6; Page 15, line 8 – page 16, line 4: Recently, many research groups have focused on determination of phase states of isoprene-derived SOA and $\alpha$-pinene-derived SOA particles using different techniques (i.e. measurements of viscosity, diffusion rate, evaporation rate, reactivity, and etc.) besides bounce experiments. Below are the relevant references. Maclean et al., and Song et al. showed that

SOA particles produced from biogenic VOC are to be liquid even at dry condition at ~290 K. Please add these references even more and compared the results in detail.

Maclean, A. M., Butenhoff, C. L., Grayson, J. W., Barsanti, K., Jimenez, J. L., and Bertram, A. K.: Mixing times of organic molecules within secondary organic aerosol particles: a global planetary boundary layer perspective, Atmos Chem Phys, 17, 13037-13048, 10.5194/acp-17-13037-2017, 2017. Song, M., Liu, P. F., Hanna, S. J., Li, Y. J., Martin, S. T., and Bertram, A. K.: Relative humidity-dependent viscosities of isoprene-derived secondary organic material and atmospheric implications for isoprene-dominant forests, Atmos Chem Phys, 15, 5145-5159, 2015.

2. Please provide the fitting parameters for Fig. 4. It would be useful for readers.
* * *

---

## Referee Comment (RC3) · Anonymous Referee #3 · 8 Sep 2018

Review of "Physical state of 2-methylbutane-1,2,3,4-tetraol in pure and internally mixed aerosols" by Lessmeier et al.

In this paper the authors 1) synthesized 2-methylbutane-1,2,3,4-tetraol, 2) measured the glass transition temperature of the tetraol, 3) determined a phase diagram for water/tetraol mixtures over the whole tropospheric temperature and RH range, 4) investigated water diffusion in the tetraol, and 5) measured the glass transition temperatures of mixtures of the tetraol with 3-methylbutante-1,2,3-tricarboxylic acid. This combined information was used to argue that isoprene-derived SOA will be a liquid in the lower troposphere and at high RH values, but a semisolid or even glassy state in the upper troposphere. The results have important implications for the growth and reactivity of atmospheric secondary organic aerosols. The paper combines first-rate physical chemistry and important atmospheric insight, and hence I strongly support this manuscript for publication in ACP. Due to the high quality of the manuscript, I have very few comments. Below are a few comments I have that the authors may want to consider before publication. Since I am not an expert on synthesis, I do not have any comments on the synthesis part of the manuscript.

Minor comments:

1) Page 5, line 3. Can the authors confirm that there was no loss of material after the DSC sample pans were sealed? In other words, did the seal prevent evaporation or water uptake? I assume this can be checked from repeated DSC measurements.

2) Figure 5. The shell of the second sphere from the left looks aqua, not light green. Should the shell be light green to be consistent with the figure caption?

3) Figure 7a. Should the annotation on the arrow be changed to "diffusion into the material (without further uptake)" or "diffusion into the particles (without further uptake)"?

4) From the water uptake experiments (e.g. Figure 7) would it be possible to extract a diffusion coefficient. This is, obviously, beyond the scope of the current paper, but I am interested in the authors' response.

5) Page 20, line 10 and Page 21, line 19-21. Can the authors comment on how the kgt they determined compares with the kgt assumed in the recent modelling study by Shiraiwa et al. 2017. Shiraiwa, M., Li, Y., Tsimpidi, A. P., Karydis, V. A., Berkemeier, T., Pandis, S. N., Lelieveld, J., Koop, T., and Poschl, U.: Global distribution of particle phase state in atmospheric secondary organic aerosols, Nature Communications, 8, 10.1038/ncomms15002, 2017.

6) Supplement, Page 4, line 13. Delete "we".

---

## Author Comment (AC1) · 10 Oct 2018

*This manuscript investigated the phase state of tetraol, which is an important secondary organic aerosol component. The authors synthesized teraol and studied the glass transition temperatures (Tg) of tetroal in pure and mixed particles with another important oxidation product, 3-MBTCA. This manuscript derived the phase diagram of water/tetraol mixtures at the atmospheric relevant conditions and determined the water activity of water/tetraol mixtures as a function of temperature and solute mass concentration. This study provides a set of valuable data for the phase state important SOA at different temperature and humidity. Due to the lack of experience on the synthesis, I was not able to provide assessment on this part. Besides that, the experimental methods are valid and the scientific approach and discussion are sound. The paper is well written and organized. I recommend it for publication for a minor revision. Please see the following comments which the authors may want to consider in the revision.*

We thank the referee for her/his positive evaluation and the comments to improve our manuscript.

*Minor comments:*

*1. P11, L17-22, It is not trivial to understand the Tg' with currently short description. It will be useful to provide details how the Tg' was determined.*

We have added a paragraph describing what Tg' is and how it can be measured (new p. 11 in marked version):

"The $T_g$' of an aqueous mixture can be measured, when in samples with high to medium water mass fractions ice crystallization is the first phase transition that occurs upon cooling. In these cases, some of the water nucleates and forms small ice crystals. Because of this ice formation the remaining solution is depleted in water and becomes enriched in tetraol until the system reaches the ice/solution equilibrium, i.e. where the current temperature matches the freezing point of the remaining solution. When the temperature is decreased further, as is done in the experiments described above, the ice crystals continue to grow and the solution becomes more and more concentrated in tetraol thus increasing in viscosity. At some point the temperature is so low and the tetraol concentration becomes so large that the remaining solution forms a glass and no further ice crystal growth occurs due to the hindered macroscopic diffusion of water molecules in the now glassy phase. This temperature is termed $T_g$' and marks the intersection of the ice melting point curve with the glass transition curve."

*It could be also useful to provide measured average values instead of Tg' is about 200 K.*

We added the exact mean Tg' values for both stereoisomers to the caption of Figure 2:

"… is depicted as the mean Tg' shifted towards the Gordon-Taylor fit (blue star) with a value of 197 K for *rac*-(2*R*,3*R*)-tetraol and 196 K for *rac*-(2*R*,3*S*)-tetraol."

*2. P13, L2-11, How does the Tg look like when using Tg(dry) from the spray-dried samples? It is suggested to derived Tg using both Tg(dry) from vacuum-dried and spray-dried samples, or provide uncertainties in Figure 2. Does the uncertainty in kGT cover this variation?*

When the $T_g$(dry) is derived from spray-dried samples the $T_g$-curves change only slightly. For the *R,S*-racemate the $k_{GT}$ value changes from 1.53 ± 0.08 to 1.68 ± 0.13, so both values agree within their respective uncertainty. For the *R,R*-racemate the $k_{GT}$ value changes from 1.58 ± 0.05 to 1.84 ± 0.15. Even though this difference is slightly outside of the respective uncertainty range, it is within the uncertainty of the atmospherically more relevant representation of the Tg curve in water activity, see Figure 4. Nevertheless, following the suggestion of the referee we now add the $k_{GT}$-values derived when using the $T_g$(dry) from the spray-dried samples in the text (new p. 13 in marked version):

"For that reason, we used the $T_g$ values of the vacuum-dried tetraols for the Gordon-Taylor analysis shown in Figure 2 to allow for a better comparability. Using the $T_g$ derived from spray-dried samples of pure tetraol changes the derived Gordon-Taylor-curve only slightly yields a $k_{GT}$ of 1.68 ± 0.13. The partial phase diagram of *rac*-(2*R*,3*S*)-tetraol (Figure 2b) is very similar to that of the diastereomeric racemate. The glass transition curve fitting results in a $k_{GT}$ of 1.58 ± 0.05 (1.84 ± 0.15 when using the $T_g$ derived from a spray-dried sample of pure tetraol)."

*L5-6, the maximum error in the water mass fraction is estimated to be 0.03 (a) and 0.05(b), do you mean in figure 2(a) and 2(b)?*

Yes. This has been corrected (new p. 13 in marked version).

*3. P14, L8-13, It will be great to provide the fitting parameters that could be directly used by the readers.*

The fitting parameters are now provided in the supporting information where the fitting procedure is explained in detail (new Table S1 and corresponding text).

*4. P16 section 3.4, is it possible to estimate the diffusion coefficient of water vapor using this water uptake/DSC experiment?*

The kinetics of the observed diffusion process of water into the pure tetraol phase depends primarily on the diffusion coefficients of water and of tetraol, both of which depend upon temperature and composition of the tetraol/water mixture. Because the DSC-measurement was done in a sealed sample pan with no further water uptake, the spatial distribution of the components, i.e. the tetraol concentration and the thickness of the shell and the core all change during the course of the experiment, and so does temperature. For these reasons the system is underdetermined for extracting a back-of-the-envelope estimate of the diffusion coefficients from this single experiment. The latter would require a composition and space-resolved model that includes a parameterization of the composition- and temperature-dependent diffusion coefficients for the full range of mixing ratios of the two components, which in turn would require several experiments at different temperatures and starting compositions, which was – as already noted by referee 3 – beyond the scope for this paper and is a topic for future studies.

---

## Author Comment (AC2) · 10 Oct 2018

*The authors describe the physical phase states of 2-Methylbutane-1,2,3,4-tetraol and mixtures of tetrol which can be a marker for isoprene-derived SOA and alpha-pinene-derived SOA. The results provide new/additional insight for the phase state of SOA particles which is still unknown in the atmospheric chemistry community. This manuscript is clear, concise, and well-written. I recommend this manuscript for publication in ACP. I have several comments that the authors should consider prior to publication.*

We thank the referee for her/his positive evaluation and the comments to improve our manuscript.

*1. Page 2, line 19 – page 3, line 6; Page 15, line 8 – page 16, line 4: Recently, many research groups have focused on determination of phase states of isoprene-derived SOA and α-pinene-derived SOA particles using different techniques (i.e. measurements of viscosity, diffusion rate, evaporation rate, reactivity, and etc.) besides bounce experiments. Below are the relevant references. Maclean et al., and Song et al. showed that SOA particles produced from biogenic VOC are to be liquid even at dry condition at 290 K. Please add these references even more and compared the results in detail.*

*Maclean, A. M., Butenhoff, C. L., Grayson, J. W., Barsanti, K., Jimenez, J. L., and Bertram, A. K.: Mixing times of organic molecules within secondary organic aerosol particles: a global planetary boundary layer perspective, Atmos Chem Phys, 17, 13037-13048, 10.5194/acp-17-13037-2017, 2017.*

*Song, M., Liu, P. F., Hanna, S. J., Li, Y. J., Martin, S. T., and Bertram, A. K.: Relative humidity-dependent viscosities of isoprene-derived secondary organic material and atmospheric implications for isoprene-dominant forests, Atmos Chem Phys, 15, 5145-5159, 2015.*

We thank the referee for pointing out to us these two references, which are now referred to at the appropriate places, i.e. on new page 2 and new page 15 in marked version.

*2. Please provide the fitting parameters for Fig. 4. It would be useful for readers.*

The fitting parameters are now provided in the supporting information where the fitting procedure is also explained in detail (new Table S1 and corresponding text).

solutions at different concentrations and heating rates. (Sorbitol is a hexane-hexol and thus structurally similar to tetraol.) From these measurements we concluded a correction value of 0.93 K that is subtracted from the measured tetraol solution ice melting points. The water activity at the heating rate-corrected ice melting points was then calculated using equation S2.

The water activities at 25°C and those at the melting points for the different concentrations was then fitted using equation S3 (Zobrist et al., 2008), whereby the fit parameters $a$, $b$ and $c$ obtained from fitting equation S1 were kept fixed

$$a_w(w_2, T) = \frac{(1 + a \cdot w_2)}{(1 + b \cdot w_2 + c \cdot w_2^2)} + (T - T^\theta) \cdot (d \cdot w_2 + e \cdot w_2^2 + f \cdot w_2^3 + g \cdot w_2^4) \quad \text{S3}$$

The last bracket in the equation is a fourth order polynomial function that is meant to describe the concentration dependence of the slope of the linear fit.

With these equations our measured glass transition temperatures at different mass fractions as well as the corresponding Gordon-Taylor fit can be converted from a mass fraction dependence to a water activity dependence. We note here that this transformation is not exact for several reasons. First we derived the temperature dependence of the water activity by fitting only two points for each concentration. Second we had to correct our ice melting points for a high heating rate which is a potential source for uncertainty. Third our data could not be fitted very well for the entire concentration range with the original fourth-order polynomial equation from Zobrist et al.: while the fourth order polynomial in equations S3 fitted the low tetraol concentration range very well, we obtained a better fit at medium tetraol concentrations with a second order polynomial (i.e. by setting parameters f and g to zero, see fitting parameter values in table S1). To take these uncertainties into account we took the following measures for the data transformation from the mass fraction dependence to the water activity dependence.

We did not transfer the actual Gordon-Taylor fit itself into the water activity regime but the 3σ range of the fit. Furthermore, we did the transformation with both the second order as well as the fourth order polynomial function and then overlapped the resulting ranges. With this procedure we derived a relatively broad glass transition range rather than an actual glass transition line. While this procedure

reduces the precision of the glass transition temperature as a function of water activity, it enhances the certainty that the glass transition will take place within the range of glass transition temperatures depicted in figure 4.

**Table S1.** Fitting parameters of equation S3 used for figure 4 in the main paper.

| rac-(2R,3R)-tetraol | | | | | | | |
|---|---|---|---|---|---|---|---|
| fitting parameter | a | b | c | d | e | f | g |
| 4th order polynomial | -0.9996 | -0.86678 | 0.06027 | 0.00973 | -0.01623 | -0.00169 | 0.00819 |
| 2nd order polynomial | -0.9996 | -0.86678 | 0.06027 | 0.0056 | -0.0056 | 0 | 0 |
| rac-(2R,3S)-tetraol | | | | | | | |
| fitting parameter | a | b | c | d | e | f | g |
| 4th order polynomial | -0.9997 | -0.86188 | 0.04763 | 0.01384 | -0.03261 | 0.02025 | -0.00148 |
| 2nd order polynomial | -0.9997 | -0.86188 | 0.04763 | 0.00631 | -0.00631 | 0 | 0 |

**Syntheses**

**General information**

*Materials used*

The following chemicals were obtained from commercial suppliers and were used as received. The quality specified by the supplier is given in parentheses: (*S*)-1,1'-bi-2-naphthol (99%), *tert*-butanol (100%), calcium chloride (85%), citric acid monohydrate (> 99.5%), 2,2-dimethoxypropane (98%), Dowex® 50 WX4 (100 - 200 mesh, Sigma-Aldrich Chemie GmbH), lipase A *Candida antarctica* immobilised on Immobead 150, recombinant from *Aspergillus oryzae* (CAL-A, ≥ 500 U/g, product nr.:41658, Sigma-Aldrich Chemie GmbH), lithium aluminium hydride (for synthesis), maleic acid (99.99%), magnesium sulfate (> 99%), methanol (100%), (*E*)-2-methylbut-2-enedioic acid (99%), (*Z*)-2-methylbut-2-enedioic acid (99+%), *N*-methylmorpholine-*N*-oxide (97%), potassium carbonate (> 99.0%), potassium osmate dihydrate (for synthesis), potassium permanganate (99%), sodium hydroxide (99%), sodium sulphite (98%), concentrated sulfuric acid (95%), toluene (99.9%), vinyl butanoate (> 98.0%.).

Tetrahydrofuran (99.7%) was distilled from sodium/benzophenone prior its use. Solvents used for work-up and chromatography were of technical grade and were distilled prior their use. Deionized water was used for the syntheses.

---

## Author Comment (AC3) · 10 Oct 2018

*Review of "Physical state of 2-methylbutane-1,2,3,4-tetraol in pure and internally mixed aerosols" by Lessmeier et al. In this paper the authors 1) synthesized 2-methylbutane-1,2,3,4-tetraol, 2) measured the glass transition temperature of the tetraol, 3) determined a phase diagram for water/tetraol mixtures over the whole tropospheric temperature and RH range, 4) investigated water diffusion in the tetraol, and 5) measured the glass transition temperatures of mixtures of the tetraol with 3-methylbutante-1,2,3-tricarboxylic acid. This combined information was used to argue that isoprene-derived SOA will be a liquid in the lower troposphere and at high RH values, but a semisolid or even glassy state in the upper troposphere. The results have important implications for the growth and reactivity of atmospheric secondary organic aerosols. The paper combines first-rate physical chemistry and important atmospheric insight, and hence I strongly support this manuscript for publication in ACP. Due to the high quality of the manuscript, I have very few comments.*
*Below are a few comments I have that the authors may want to consider before publication. Since I am not an expert on synthesis, I do not have any comments on the synthesis part of the manuscript.*

We thank the referee for her/his positive evaluation and the comments to improve our manuscript.

*Minor comments:*

*1) Page 5, line 3. Can the authors confirm that there was no loss of material after the DSC sample pans were sealed? In other words, did the seal prevent evaporation or water uptake? I assume this can be checked from repeated DSC measurements.*

The DSC sample pans are closed with an aluminum lid and then sealed with a crimper press, which is why they are called "hermetic pans". This sealing indeed allows no water evaporation or uptake which was confirmed by repeated DSC measurements and by checking the sample mass before and after the measurements via weighing. We have added the term "hermetically sealed" to that sentence.

*2) Figure 5. The shell of the second sphere from the left looks aqua, not light green. Should the shell be light green to be consistent with the figure caption?*

Yes, the referee is correct, thanks for pointing this out to us. We have revised the figure accordingly and also slightly modified the figure caption.

*3) Figure 7a. Should the annotation on the arrow be changed to "diffusion into the material (without further uptake)" or "diffusion into the particles (without further uptake)"?*

Yes, this is a very good and valid point: "into the material" is indeed a better description and we have changed the text accordingly, see new Fig.7a.

*4) From the water uptake experiments (e.g. Figure 7) would it be possible to extract a diffusion coefficient. This is, obviously, beyond the scope of the current paper, but I am interested in the authors' response.*

The kinetics of the observed diffusion process of water into the pure tetraol phase depends primarily on the diffusion coefficients of water and of tetraol, both of which depend upon temperature and composition of the tetraol/water mixture. Because the DSC-measurement was done in a sealed sample pan with no further water uptake, the spatial distribution of the components, i.e. the tetraol concentration and the thickness of the shell and the core all change during the course of the experiment, and so does temperature. For these reasons the system is underdetermined for extracting a back-of-the-envelope estimate of the diffusion coefficients from this single experiment. The latter would require a composition and space-resolved model that includes a parameterization of the composition- and temperature-dependent diffusion coefficients for the full range of mixing ratios of the two components, which in turn would require several experiments at different temperatures and starting compositions, which was – as already noted by the referee – beyond the scope for this paper and is a topic for future studies.

*5) Page 20, line 10 and Page 21, line 19-21. Can the authors comment on how the kgt they determined compares with the kgt assumed in the recent modelling study by Shiraiwa et al. 2017. Shiraiwa, M., Li, Y., Tsimpidi, A. P., Karydis, V. A., Berkemeier, T., Pandis, S. N., Lelieveld, J., Koop, T., and Poschl, U.: Global distribution of particle phase state in atmospheric secondary organic aerosols, Nature Communications, 8, 10.1038/ncomms15002, 2017.*

We have added a short paragraph discussing this issue on new p. 22/23 in marked version:

"We note that the value of $k_{GT}$ = 3.79 ±0.15 of the binary tetraol/3-MBTCA mixture determined here differs significantly from the value of $k_{GT}$ = 1 assumed for multicomponent SOA mixtures in a recent modeling study (Shiraiwa et al., 2017). We also note that to our knowledge this tetraol/3-MBTCA mixture is the second binary mixture of individual SOA compounds investigated so far, the other being pinonic acid/3-MBTCA (Dette et al., 2014). Hence, it would be premature to conclude that these two binary systems are representative for the entire variety of atmospheric binary mixtures, which may also exhibit $k_{GT}$-values smaller than one. Therefore, as the different $k_{GT}$ values may compensate each other in multicomponent mixtures a mean $k_{GT}$ of one is not implausible, although our present study may be a first hint that $k_{GT}$ may be slightly larger than 1."

*6) Supplement, Page 4, line 13. Delete "we".*
Thanks, "we" has been deleted.